# The Interplay between Inflammation, Anti-Angiogenic Agents, and Immune Checkpoint Inhibitors: Perspectives for Renal Cell Cancer Treatment

**DOI:** 10.3390/cancers11121935

**Published:** 2019-12-04

**Authors:** Nicole Brighi, Alberto Farolfi, Vincenza Conteduca, Giorgia Gurioli, Stefania Gargiulo, Valentina Gallà, Giuseppe Schepisi, Cristian Lolli, Chiara Casadei, Ugo De Giorgi

**Affiliations:** 1Medical Oncology Department, Istituto Scientifico Romagnolo per lo Studio e la Cura dei Tumori (IRST) IRCCS, 47014 Meldola, Italy; nicolebrighi@hotmail.com (N.B.); vincenza.conteduca@irst.emr.it (V.C.); giuseppe.schepisi@irst.emr.it (G.S.); cristian.lolli@irst.emr.it (C.L.); chiara.casadei@irst.emr.it (C.C.); ugo.degiorgi@irst.emr.it (U.D.G.); 2Bioscience Laboratory, Istituto Scientifico Romagnolo per lo Studio e la Cura dei Tumori (IRST) IRCCS, 47014 Meldola, Italy; giorgia.gurioli@irst.emr.it (G.G.); stefania.gargiulo.94@gmail.com (S.G.); 3Unit of Biostatistics and Clinical Trials, Istituto Scientifico Romagnolo per lo Studio e la Cura dei Tumori (IRST) IRCCS, 47014 Meldola, Italy; valentina.galla@irst.emr.it

**Keywords:** kidney cancer, immunotherapy, renal cell, inflammation markers, programmed death-ligand 1, immune checkpoint inhibitors, prognostic factors, predictive factors

## Abstract

Treatment options for metastatic renal cell carcinoma (RCC) have been expanding in the last years, from the consolidation of several anti-angiogenic agents to the approval of immune checkpoint inhibitors (ICIs). The rationale for the use of immunomodulating agents derived from the observation that RCC usually shows a diffuse immune-cell infiltrate. ICIs target Cytotoxic T Lymphocytes Antigen 4 (CTLA-4), programmed death 1 (PD-1), or its ligand (PD-L1), showing promising therapeutic efficacy in RCC. PD-L1 expression is associated with poor prognosis; however, its predictive role remains debated. In fact, ICIs may be a valid option even for PD-L1 negative patients. The establishment of valid predictors of treatment response to available therapeutic options is advocated to identify those patients who could benefit from these agents. Both local and systemic inflammation contribute to tumorigenesis and development of cancer. The interplay of tumor-immune status and of cancer-related systemic inflammation is pivotal for ICI-treatment outcome, but there is an unmet need for a more precise characterization. To date, little is known on the role of inflammation markers on PD-1 blockade in RCC. In this paper, we review the current knowledge on the interplay between inflammation markers, PD-1 axis, and anti-angiogenic agents in RCC, focusing on biological rationale, implications for treatment, and possible future perspectives.

## 1. Introduction

Renal cell carcinoma (RCC) is the seventh most common type of cancer in men and the tenth in women in Western countries [1,2]. RCC incidence has been increasing in the last 30 years, at an annual rate of around 3%, but the figures are recently showing a tendency of plateauing [3]. At the time of diagnosis, 25% to 30% of patients present with metastatic disease associated with high mortality. However, when all stages of RCC are considered, mortality rates seem to have leveled [4]. In fact, the widespread use of noninvasive radiological techniques leads to frequent incidental detection of early and small kidney tumors, which are potentially curable.

For many years, treatments for advanced RCC were limited to interferon α (IFNα) and interleukin (IL)-2. After the cytokine era, two more categories of drugs became available, namely anti-angiogenic agents and mammalian target of rapamycin (mTOR) inhibitors. In the last years, immune-checkpoint inhibitors (ICIs) obtained indication at first as second-line treatment and are now available also as first-line treatment in metastatic RCC.

In this paper, we review the current knowledge on the interaction of inflammation and the PD-L1/PD-L1 axis in RCC, focusing on their possible role as prognostic and predictive factors in patients affected by these tumors and treated with ICIs or anti-angiogenic agents.

## 2. Anti-Angiogenic Agents in RCC Treatment

Anti-angiogenic agents, such as various tyrosine kinase inhibitors (TKIs) (i.e., sunitinib, axitinib, sorafenib, pazopanib, and lenvatinib), target multiple receptors for platelet-derived growth factor (PDGF-Rs) and vascular endothelial growth factor receptors (VEGFRs), which play a role in both tumor angiogenesis and tumor-cell proliferation. Similarly, bevacizumab, a recombinant humanized monoclonal antibody, blocks angiogenesis by inhibiting vascular endothelial growth factor A (VEGF-A). Also, the mesenchymal–epithelial transition (MET) and multityrosine kinases inhibitor cabozantinib is currently used in advanced RCC.

The use of these drugs resulted in improved outcomes, particularly for overall survival (OS) (sunitinib, pazopanib, and cabozantinib) and for progression-free survival (PFS) (sunitinib, axitinib, cabozantinib, sorafenib, and pazopanib) [5,6,7,8,9,10,11,12].

## 3. Immune Checkpoint Inhibitors in RCC Treatment

In recent years, therapeutic options for RCC have expanded, and the use of ICIs, has been approved. Nivolumab, targeting programmed-death receptor 1 (PD-1), and ipilimumab, directed against cytotoxic T lymphocytes antigen 4 (CTLA-4), are currently considered standard treatment options for RCC. The rationale for the use of these drugs lies in the inhibitory role on specific pathways related to the immune response, frequently hyperactivated by tumor-cell interaction. By inhibiting these pathways, ICIs reactivate an immune response against tumor cells. The high mutation load typical of RCC probably correlates with a high antigen expression and has led to the testing of these drugs at different stages of the disease. CheckMate 025 was a large phase III clinical trial, comparing nivolumab (PD-1 inhibitor) to everolimus in patients with locally advanced or metastatic RCC, progressed after treatment with at least one VEGF/VEGFR inhibitor. The study showed an OS benefit in patients treated with nivolumab. Furthermore, the immunotherapy-treated cohort had a higher overall response rate (ORR) compared to everolimus, with a considerable rate of long-lasting responses [13]. Due to these satisfactory results, ICIs are being tested in earlier settings (adjuvant and neo-adjuvant) and are now also available as first-line treatment [14,15].

In fact, another large phase III study has demonstrated that the combination of ipilimumab and nivolumab was superior to sunitinib in intermediate- and poor-risk patients when used as first-line treatment. In this population, the association of the two ICIs improved OS, as well as response rate, with a complete response rate of about 10% [15].

Furthermore, following the mounting evidence of the interaction between angiogenesis and immune escape, several trials have been designed and conducted to evaluate the role of the association of ICIs with antiangiogenic agents as first-line treatment (see Table 1) or further.

For example, the IMmotion150 study, a phase II trial, evaluated the use of atezolizumab (an anti-PD-L1 inhibitor) plus bevacizumab compared to atezolizumab alone or sunitinib as first-line treatment for locally advanced and metastatic RCC. The study showed that patients treated with the association of atezolizumab and bevacizumab had a longer PFS compared to atezolizumab (6.1 months) and sunitinib arms; furthermore, a higher percentage ORR was reported in the combination arm. Interesting results were observed in patients with PD-L1 positive expression (≥1%), achieving a longer PFS (14.7 months) and higher ORR (46%) in the atezolizumab monotherapy arm [16].

Two recent phase III trials (JAVELIN Renal 101 and KEYNOTE-426) investigated the role of the association of ICIs and TKIs and showed better outcomes in patients treated respectively with the association of avelumab plus axitinib or pembrolizumab plus axitinib compared to sunitinib in previously untreated advanced RCC [17,18].

COSMIC-313 (NCT 03937219) is a multicenter, randomized, controlled phase III trial that evaluates the combination of cabozantinib/placebo plus nivolumab and ipilimumab in previously untreated intermediate- and poor-risk RCC.

The interplay of inflammatory mediators and pathways related to immune response is extremely complex, and its role is pivotal both for RCC tumorigenesis and for treatment response.

## 4. Inflammation and Cancer and the PD-1/PD-L1 Axis

The potential relation between cancer and inflammation was originally proposed by Virchow in the 19th century [27]. In the following years, the pivotal role of inflammation in mediating tumorigenesis, progression, and metastasis of cancer has progressively been unraveled and recognized [28,29].

It has been observed that systemic inflammation damages the immune response, allowing tumor cells to escape from immune surveillance. Cancer immune surveillance is a fundamental host-defense process to inhibit carcinogenesis. However, cancer cells can progressively adapt to escape from immune-mediated rejection, through the process of “immune editing”, resulting from a selective pressure on the tumor microenvironment, leading to tumor progression [30].

The role of systemic inflammation, and particularly of all the leucocytes cells, in aiding tumoral immune escape is mediated by the inhibition of apoptosis, promotion of genomic instability and tumoral invasion, angiogenesis, and metastatic spread through different processes [27,28,31].

Neutrophils can be recruited by the tumor through the production of cancer-related chemokines and cytokines (e.g., IL-6 and TNF). Neutrophils are involved in the proliferation, invasion, and metastatic spread of tumor cells, also inducing drug resistance [30,32].

A lower level of lymphocytes, which is frequently observed in systemic inflammation, is frequently related to low levels of CD4+ T cells, resulting in less-effective immune surveillance. On the other hand, tumor-associated macrophages (TAMs), deriving from monocytes, can promote invasion, proliferation, and angiogenesis of tumor cells, thus favoring cancer spread and metastases formation [28,30].

Cyclooxygenase-2 (COX-2) is involved in the conversion of arachidonic acid to prostaglandin H2, an important precursor of prostacyclin, which is expressed widely in inflammation and in tumors. In RCC, COX-2 expression is present in the majority of the tumors and correlates with a worse stage and grade and poorer outcomes [33,34,35].

Epidemiological data demonstrate that the use of nonsteroidal anti-inflammatory drugs (NSAIDs), including long-term use of aspirin, decreases incidence, metastasis, and mortality risk in several cancers [36,37,38,39,40]. The antitumoral effect of NSAIDs is thought to be related mainly to the inhibition of COX-2, but COX-independent mechanisms have also been observed [41]. NSAIDs have been shown to inhibit tumor growth by inducing cancer-cell apoptosis and inhibiting the Wnt/β-catenin signaling pathway [42]. NSAIDs may therefore inhibit the development of early malignant lesions and cause regression of tumors in animal models of colorectal cancers and a decrease in recurrence rates of adenomas [43,44,45]. In RCC, NSAIDs (but not aspirin) have been implicated as a risk factor for RCC development [46,47]. In the setting of metastatic RCC, a retrospective analysis on 4736 patients included in several phase II and phase III trials concluded that NSAIDs do not confer a survival advantage in mRCC patients [48].

Taking into account the paramount importance of inflammation in cancer development, the role of IL-1 as potential mediator for tumoral angiogenesis and metastatic spread is being investigated in various preclinical models. Two recent studies showed that IL-1b promotes the stem-cell properties of gastric cancer cells, through activation of the phosphoinositide 3-kinase pathway [49] and the IL-1b/IL-6 network is highly expressed in human colorectal cancer, reinforcing the possible correlation of the inflammatory mediators with cancer progression [50].

Canakinumab is a human monoclonal antibody against interleukin-1b. Canakinumab has been proven to significantly reduce atherosclerosis and other cardiometabolic diseases related to inflammation (diabetes, stroke, and chronic kidney disease) [51]. Interestingly, a significant reduction in the incidence of lung cancer in patients treated with higher doses of canakinumab was observed. However, no significant decrease in the rate of other primary cancers was observed in the canakinumab group, as compared with the placebo.

Currently, to investigate the role of this drug on renal cell cancer, an early phase I trial (SPARC-1, NCT04028245) is enrolling patients to be treated with the association of canakinumab and spartalizumab (an anti-PD1 monoclonal antibody) as neoadjuvant treatment before radical nephrectomy.

The role of PD-1/PD-L1 axis in mediating tumor escape from the immune system has been widely investigated in the last years. Under normal conditions, the immune system can recognize cancer cells and induce apoptosis mediated by T-cell activation. The PD-1/PD-L1 pathway is an adaptive immune resistance mechanism used by cancer cells in response to the host immune-related antitumor activity. PD-L1 is overexpressed in tumor cells or in its microenvironment, and it binds to PD-1 receptors on the activated T cells, resulting in the inhibition of cytotoxic T cells and tumor escape.

Human PD-1 is a type I transmembrane glycoprotein of the CD28/CTLA-4 immune checkpoint receptor family; its ligands are PD-L1 and PD-L2 [52,53]. PD-1 is expressed in hematopoietic tissues and cells, including T cells, B cells, NK cells, monocytes/macrophages, and dendritic cells [54]. PD-1 expression is rapidly induced consequently to T- or B-cell antigen stimulation, or upon lymphoid-cell activation conditions. The expression pattern of PD-L1 is wider, displaying both constitutive and inducible expression in lymphoid, myeloid, and endothelial cells [55]. PD-L1 expression is high in many human cancers, both in the tumor-infiltrating immune cells and in the tumor cells [53,56]. In RCC patients, PD-1 is highly expressed on the surface of both activated tumor-infiltrating immune cells and peripheral blood cells [57].

The main effect of PD-1/PD-L1 interaction at the base of immune evasion is the negative signaling on the antigen receptor complexes, resulting in the trigger of the reversal of CD3-complex tyrosine phosphorylation and the decline in cytokine production and lymphocyte proliferation [58]. In addition, PD-L1 also plays protumorigenic roles in cancer cells by binding to its receptors in hematopoietic cells, which results in activation of proliferative- and survival-signaling pathways, leading to subsequent tumor progression [59].

PD 1/PD-L1 and CTLA 4 inhibitors can target specific pathways related to immune-response hyperactivated by tumor-cell interaction. By inhibition of these targets, ICIs could reactivate a specific immune response against tumor cells. Agents targeting the PD-1/PD-L1 axis increase the proliferation and cytolytic activity of T cells, resulting in durable ORR [53].

It has been demonstrated that there is a synergistic effect of ICIs and anti-angiogenic agents (Figure 1). Besides their direct inhibitory effect on angiogenesis, anti-angiogenic agents can reverse tumor-related immune suppression through several mechanisms, such as the decrease of immunosuppressive cells (MDSCs, regulatory T cells) and cytokines (TGFβ and IL-10) and the direct inhibiting interaction on PD-1 on T cells [60]. Thus, the use of a combination of these agents and ICIs has a strong biological rationale and is currently the object of many clinical and preclinical research studies.

## 5. Inflammation as Prognostic Factor

Prognostic factors for localized RCC include anatomical, histological, clinical, and molecular features. Tumor stage (as per TNM staging) and Fuhrman nuclear grade are the strongest independent prognostic factors for localized RCC. However, the use of integrated systems (such as the UCLA Integrated Staging System (UISS); the Stage, Size, Grade and Necrosis (SSIGN) system; or other nomograms) combining multiple independent prognostic factors results in higher accuracy [61,62,63,64]. These tools have been widely used in clinical practice, to guide decision making. Several molecular and genetic markers have been investigated as potential prognostic factors for RCC: Von Hippel Lindau (VHL) gene alterations, hypoxia-induced factor 1 alpha (HIF-1a), mTOR, ribosomal protein S6 and phosphatase PTEN, Ki-67, levels of carbonic anhydrase 9 (CAIX), and matrix metalloproteinase 2 and 9 [64]. However, their role on determining prognosis is still controversial and only for investigational use. In the metastatic setting, the classical anatomical factors (stage, size, perinephric fat, and venous or adrenal invasion) used as prognostic factors in localized RCC have a very limited prognostic role. In fact, the prognostic role of the primary tumor on prognosis disappears when the tumor becomes metastatic. Some metastases features have been proved to be reliable prognostic factors. For example, the resectability of metastases is currently considered an independent prognostic factor, regardless of the anatomic site [65,66]. The presence of multiple resectable pulmonary lesions with nodal involvement is associated with worse prognosis. Bone and spinal metastases are associated with poor outcomes, similarly to the presence of multiple brain metastases [67,68,69].

Regarding the role of histology as prognostic factor in metastatic RCC, the histological subtype and the presence of sarcomatoid component have prognostic significance. Sarcomatoid differentiation is clearly associated with very poor prognosis [70].

As for clinical criteria, performance status (assessed either by Karnofsky index or Eastern Cooperative Oncology Group (ECOG) scale) is the most important clinical prognostic factor in metastatic RCC, regardless of the class of the type of treatment [71,72,73]. Other significant clinical prognostic variables are presence or absence of previous nephrectomy, time from nephrectomy to treatment, and time from diagnosis to treatment [64,73].

Additionally, low hemoglobin, elevated lactate dehydrogenase, corrected serum calcium, and inflammatory markers have been historically described as significant prognosticators [74,75,76]. The International Metastatic Renal-Cell Carcinoma Database Consortium (IMDC) model, one of the most-used indexes in clinical practice, efficaciously stratifies patients into three risk groups combining several clinical parameters (anemia, neutrophilia, thrombocytosis, hypercalcemia, Karnofsky performance status and time from diagnosis to treatment) [77].

Increasing evidence is showing the existence of a complex interplay between several inflammation factors and the prognosis of patients with various types of cancers, including RCC [10,30]. The biological rationale lies in the concept that local immune response and systemic inflammation play an important role in the initiation, development, and progression of cancer. Inflammatory cells, such as neutrophils, monocytes, and lymphocytes, promote the intravasation of neoplastic cells in the circulation system, allowing the growth of distant metastases. This mechanism is thought to be one of the reasons contributing to patients’ poor outcomes [78,79]. In clinical practice, systemic inflammation is easily evaluated by peripheral blood counts of immune and inflammatory cells and acute-phase proteins such as C-reactive protein (CRP). Several inflammation markers or inflammation indexes have been identified. Inflammation indexes combine and integrate conventional inflammatory parameters and have been proved to be potential prognostic values in several studies focusing on various types of cancer [79,80,81,82,83].

Serum CRP is produced in hepatocytes; its production is regulated by inflammation-associated cytokines (IL-6 and IL-1b). These cytokines are produced in various cells, including inflammatory cells and cancer cells. Thus, serum CRP concentration might be elevated due to hepatic stimulation by cancer-cell-derived inflammatory cytokines [84].

Elevation of CRP is associated with poorer survival in patients with cancer [85,86]. CRP dosing is easily reproducible, inexpensive, and has good sensitivity. Several studies have shown that CRP can be considered a good prognostic factors both in localized and metastatic RCC, with a significant association between high CRP and worse outcomes [87,88,89,90]. One of the greatest limits of CRP as a prognosticator is the lack of a specific cutoff, due to the different values used in the available studies.

The most promising indexes identified so far are the neutrophil to lymphocyte ratio (NLR), the platelet to lymphocyte ratio (PLR), the prognostic nutritional index (PNI), the systemic immune-inflammation index (SII), and the systemic inflammation response index (SIRI).

Neutrophils, lymphocytes, and macrophages have been implicated in promoting neoplastic angiogenesis and metastases diffusion, but they are also thought to be involved in the formation of premetastatic niches and in primary and acquired drug resistance [91].

NLR, defined as the absolute neutrophil count divided by the absolute lymphocyte count, is probably the most-studied prognostic index. An increased NLR is associated with poor prognosis in several tumors such as breast, lung, pancreatic, colorectal, gastric, urothelial, prostate, ovarian, and kidney cancers [91,92,93]. Neutrophils are known to be able to aid the proliferation and survival of malignant cells, promoting angiogenesis and metastasis. Conversely, lymphocytes suppress tumor growth and invasion through their cytolytic activity. Taken together, patients with high NLR have a relative lymphopenia, and this may result in a poor immune response and worse outcomes. In RCC, lymphopenia in preoperative blood count has been associated with poor prognosis, and in elderly patients with RCC treated with sunitinib [94,95]. A large number of study and several meta-analyses have reported that increased NLR is associated with poor prognosis in RCC [93,96,97,98,99,100,101,102,103,104,105,106,107,108,109,110,111].

Baum et al. reported that NLR ≥ 4 was associated with a shorter OS as compared to RCC patients with NLR < 4 [112].

SII (defined as platelets*neutrophils/lymphocytes) combines these three parameters and has already been proved to be significantly associated with prognosis in hepatocellular carcinoma and in colorectal cancer [85,113]. Due to high levels of neutrophils and platelets and low levels of lymphocytes, a higher SII usually indicates a stronger inflammatory and a weaker immune response in patients. It may be associated with invasion and metastasis of tumor cells, and hence leads to poor survival.

In a recent retrospective series by Lolli et al., SII was identified as being a reliable parameter both as prognostic and predictive factor in RCC patients treated with sunitinib. Baseline SII values were independent factors for PFS and OS [114]. However, not all studies agree on the prognostic value of SII. To shed light on this issue, a recent metanalysis investigated the significance of SII in determining the prognosis of patients affected by several kinds of cancers [115]. The metanalysis confirmed that elevated SII indicates poor prognosis and may be considered a cost-effective prognostic biomarker. However, of the 15 papers included in the analysis, only one was based on RCC patients [114]. Therefore, the clinical significance of SII as prognostic factor should be further evaluated. Glasgow Prognostic Score (GPS), or its modified version (mGPS), is an index based on the combination of serum CRP and albumin. GPS and mGPS are considered a measure of systemic inflammation, representing the immune response and nutritional status of patients. Many studies have shown the independent prognostic value of GPS/mGPS in various types of cancers [116,117,118]. GPS/mGPS has the advantage of using a specific cutoff value, allowing comparison among studies.

In RCC, metastatic patients are reported as presenting an elevated GPS/mGPS compared to non-metastatic patients. Several studies showed a significant association between elevated GPS/mGPS and worse prognosis [119,120,121,122,123].

Platelet count has also been shown to be related to prognosis in RCC [77,124]. Platelet-to-lymphocyte ratio (PLR) is defined as the platelet count divided by the lymphocyte count. It is an easily acquirable and cheap marker. Higher PLR was significantly associated with worse outcomes in different types of cancers [99,125,126]. However, most studies regarding RCC failed to confirm the prognostic value of PLR in this setting [125,127,128].

PNI has been introduced as a simple and reproducible biomarker, reflecting the nutritional and immunological status of cancer patients. PNI is a combination of serum albumin and peripheral blood lymphocyte count. High values of PNI are associated with a good prognosis in cancer patients [84]. Studies evaluating the prognostic value of PNI in patients with RCC are few but confirm these results [129,130,131,132,133,134,135].

Recently, the determination of SIRI index (defined as neutrophils x monocytes/lymphocytes) has been determined to be a reliable prognostic factor in different kinds of tumors [136,137,138].

Besides aiding immune surveillance evasion, metastatic adhesion of cancer cells, and tumor angiogenesis by the production of proangiogenic factors, neutrophils can secrete large amounts of reactive oxygen species and nitric oxide, leading to T-cell disorders. Tumor-associated macrophages derive from circulating monocytes, which can be recruited to the tumor tissue and promote the growth and migration of the tumor. Another role of tumor-associated macrophages is to induce the apoptosis of activated T cells, resulting in the formation of new tumor vessels. Peripheral monocyte counts may reflect the level of tumor-associated macrophages, and higher monocyte counts are considered negative markers for tumors. Lymphocytes are crucial components of the immune system, serving as the main defense against cancer cells. They inhibit tumor progression by releasing cytokines such as interferon-γ and tumor necrosis factor-α [139]. Downregulation of peripheral lymphocytes causes an impairment of anticancer immunity and increases tumor-cell dissemination. The index SIRI combines these aforementioned cell types and reflects the complex interplay between immune and inflammatory cells in the tumor microenvironment [79].

In RCC, the potential prognostic value of SIRI has been evaluated in several studies, but the results are largely controversial.

Chen et al. investigated the role of SIRI in 414 patients and with an independent validation cohort of 168 patients with localized or locally advanced RCC who received radical or partial nephrectomy, suggesting that SIRI might be a better prognostic predictor than PLR, NLR, MLR, and MSKCC score [30].

Overall, inflammation markers and derived indexes as prognostic factors are easily measurable, reproducible, cheap, and thus advantageous.

However, some limitations need to be taken into account. One of these includes cutoff values that have not been clearly determined yet, thus strongly limiting their usefulness in the management of patients. Further studies are indeed needed to allow the use of these tools in clinical practice.

Several recent studies and meta-analyses have shown that PD-L1 expression correlates with clinical–pathological prognostic factors in RCC, such as the WHO/ISUP grade, presence of necrosis and sarcomatoid features, tumor size, and TNM stage [140,141].

Furthermore, many studies in RCC have suggested that patients with intratumoral high PD-L1 expression exhibited aggressive behavior and are related to poor outcomes, with an increased risk of cancer-related death [142,143,144,145,146].

However, it is known that RCC is a very heterogeneous disease, and PD-L1 expression might vary within primary tumor and between primary neoplasm and metastases, greatly limiting the value of this biomarker [147]. Another pitfall related to PD-L1 expression evaluation is the use of various antibodies in clinical practice that causes a lack of standardization and, consequently, difficulties in comparing studies’ results. In Table 2, we report phase II and phase III trials evaluating various ICI regimens, along with the characteristics of the several PD-L1 antibody clones and different cutoffs used.

In addition to PD-L1 expression, identification of the lymphocyte density in the tumor microenvironment as a prognostic biomarker could facilitate detecting patients who could benefit from the checkpoint blockade [148].

## 6. Inflammation as Predictive Factor

The clinical scenario of the treatment of RCC has changed dramatically in recent years due to the development, after the cytokine era, of molecular-targeted agents at first and subsequently of ICIs.

The availability of effective treatments represents a challenge for clinicians, who need valid tools to predict response to therapy and patients’ prognosis.

However, the search for predictive markers has not led to satisfactory results yet, and robust biomarkers for the several available classes of treatments are still lacking.

When cytokine-based immunotherapy was the only available treatment for RCC, the French group of Immunotherapy validated a prognostic model based on performance status, metastases features, time from diagnosis to treatment, hemoglobin level, neutrophil absolute count, and other biological parameters related to inflammation. This model was developed to predict the outcome of RCC patients after cytokine-based treatment, stratifying three prognostic groups (good, intermediate, and poor risk) with different median OS [149].

In the era of molecular-targeted treatments, several studies reported no association of various biomarkers with the outcome of patients treated with sunitinib, pazopanib, and everolimus. Similarly, cMET expression did not result as a good predictor for cabozantinib treatment [150].

Ongoing studies are evaluating the role of several potential markers of the VHL pathway or the mTOR pathway, such as the VEGF gene family, CAIX, VEGFRs, PDGFRs, VHL and pAkt, PTEN, p27, and pS6 [151,152,153]. However, to date, no valid biomarker has been identified in this setting.

As for the use of predictive systems combining several independent variables (such as the classification of the French group of immunotherapy for predicting outcomes after cytokines treatment, or the model by Choueiri et al., focusing on prognosis after anti-angiogenic treatments), the predictive accuracy of available models in the metastatic setting is inferior to those developed for localized RCC [72,149].

Motzer et al. developed a nomogram to predict PFS after first-line treatment with sunitinib. The model included several clinical parameters, such as hemoglobin levels, platelet count > 400,000/uL, corrected serum calcium levels, alkaline phosphatase and lactate dehydrogenase levels, ECOG performance, prior nephrectomy, number and site of metastases, and time from diagnosis to treatment [73].

Shin et al. demonstrated that PD-L1 expression is significantly related to poor response to VEGF-TKI; in addition, PD-L1 is independently associated with shorter survival in metastatic RCC patients after VEGF-TKI treatment [154]. Hara et al. also showed that positive expression of immune-checkpoint-associated molecules, including PD-1, PD-L1, and PD-L2, is related to poor outcomes in metastatic RCC patients who received TKIs as first-line systemic therapy [140]. A retrospective study by Ueda et al. reported that PD-1 expression is not only a prognostic indicator for poor OS in patients with metastatic RCC receiving molecular targeted therapies [141].

In the last years, with the advent of the ICI era, systemic inflammatory status has been proved to be associated with clinical outcomes in the treatment for RCC [155,156,157].

Some studies have tried to identify the role of CRP in patients treated with molecular-targeted agents: Patient with the deepest decreases of CRP after treatment had a significantly better outcome [89,90]. The NLR has also been evaluated as a possible predictive marker for molecular-targeted agents. Park et al. showed that a lower post-treatment NLR and larger reduction in NLR after sunitinib treatment were significantly associated with better responses in patients with advanced RCC [107].

If inflammation-related biomarkers may be considered good prognostic factors, their role as predictive factor has not been established yet, particularly in the setting of ICIs. In fact, studies on the correlation of inflammatory markers and response to immunotherapy are few, mostly with a retrospective nature and with very small cohorts of patients.

A recent meta-analysis aimed to study the effectiveness of NLR as a predictive factor in patients receiving ICIs. Although the meta-analysis included relatively few studies on different cancer types, the results suggest that NLR is a potentially useful predictive tool [158].

In a retrospective series on 42 patients by Jeyakumar et al., the role of NLR as predictor of ORR, PFS, and OS in mRCC patients treated with ICIs was evaluated. The authors demonstrated that baseline NLR < 3 was an independent predictor of longer PFS and OS in patients treated with ICIs [159].

In another recent series by the same authors, NLR > 4 was associated with shorter OS and PFS in 57 patients receiving ICIs for RCC or urothelial carcinomas [160].

Bilen et al., in a retrospective series on 38 patients, observed that low baseline NLR was associated with longer PFS and OS in patients with metastatic RCC who received nivolumab therapy. Notably, they used a cutoff value of 5.5 for NLR, which was different from the prior studies [161].

A study by Lalani et al. on 142 patients treated with ICIs in any line confirmed that patients with a higher baseline NLR had a worse prognosis and reported that NLR decrease ≥25% during ICI treatment was significantly associated with improved outcomes (both PFS and OS) [162].

These results suggest that NLR can be considered an attractive, cost-effective biomarker that can easily be measured from routine laboratory data.

However, due to the retrospective nature and the small cohorts of patients included in these studies, the results need to be interpreted with caution and need validation, possibly through larger population and prospective trials. It is also clear that a valid cutoff value has not been identified yet and the values defining elevated NLR are different in each of these studies.

The role of predictive factors of ICI treatment in metastatic RCC patients was evaluated in a large Italian retrospective study, collecting data from a prospective cohort of 389 patients enrolled in the Italian Expanded Access Program (EAP) who were treated with nivolumab [162].

The authors report that patients with a high SII (≥1375) had a significantly shorter OS and that, at multivariate analysis, SII seemed to be superior to NLR in predicting outcomes. Furthermore, SII changes at three months after treatment started predicted survival [163].

To our knowledge, studies on other inflammation-related biomarkers (e.g., PLR, SIRI, PNI, and GPS) during ICI treatments are lacking but eagerly awaited.

Tumor-associated PD-L1 expression has been proposed as a potential predictive biomarker for PD-1 pathway expression in many cancer types, although it is limited by the abovementioned pitfalls. A recent meta-analysis evaluated PD-1 and PD-L1 inhibitors’ efficacy compared to other treatments in patients with different tumors (mainly lung cancer, but also RCC, melanoma, head/neck cancer, and urothelial cancer) classified in two groups according to PD-L1 expression. PD-L1 expression resulted as a very unsatisfactory predictor because ICIs were beneficial in both groups [164].

The CheckMate 025 phase III trial evaluated nivolumab (3 mg/kg every 2 weeks) versus everolimus (10 mg daily). The trial included 821 patients with previously treated metastatic RCC [13].

The median OS was 25 months in the nivolumab group and 19.6 months in the everolimus group; PFS was not different among the two groups.

Interestingly, PD-L1-positive patients had worse outcomes in both treatment arms. Moreover, nivolumab showed clinical benefit both in PD-L1-positive and PD-L1-negative patients.

On the opposite side, patients with positive PD-L1 expression were reported to have more clinical benefit from ICIs both in the Immotion150 trial and Checkmate 214 [15,16].

The first study evaluated the use of atezolizumab plus bevacizumab versus atezolizumab plus sunitinib in untreated metastatic RCC [16].

In Checkmate, 214 treatment-naive advanced or metastatic RCC were treated with nivolumab plus ipilimumab or sunitinib [15]. In the combination arm, patients reported higher ORR in intermediate/poor risk patients. Of note, patients with intermediate/poor risk disease and PD-L1 expression ≥1% had higher ORR and PFS when treated with an ICI combination compared to sunitinib, while patients with favorable category of risk (showing lower PD-L1 expression) displayed a longer PFS and a higher ORR if treated with sunitinib.

Two recent phase III trials have focused on the association between ICIs and axitinib, exploring also the possible role of PD-L1 on treatment outcomes [17,18]. Axitinib showed clinical activity and an acceptable safety profile as first-line treatment of metastatic RCC in a phase III trial, in which it was compared to sorafenib [165].

The phase III JAVELIN Renal 101 trial reported that patients with PD-L1 positive, advanced RCC receiving first-line avelumab plus axitinib had significantly better outcomes than those treated with sunitinib. Interestingly, better PFS and ORR were observed in the avelumab plus axitinib, regardless of PD-L1 expression [17].

In the KEYNOTE-426 3 trial, treatment with pembrolizumab plus axitinib resulted in significantly longer OS and PFS, as well as a higher ORR, compared to sunitinib in previously untreated patients with advanced RCC. Similarly to the JAVELIN Renal 101 trial results [17], the benefit of pembrolizumab plus axitinib was observed in both PD-L1 expression subgroups (<1 and ≥1) and defined according to the combined positive score (defined as the number of PD-L1–positive cells, such as tumor cells, lymphocytes, and macrophages, divided by the total number of tumor cells, multiplied by 100) [18].

In a recent study by Tatli-Dogan et al., PD-L1 expression was reported to be associated with high HIF-2α expression (suggesting a possible regulation of PD-L1 by HIF-2α) and dense lymphocytic infiltration. The authors suggest that these parameters could be used as predictive factors and these patients could benefit from PD-L1-targeted therapy [148].

Particular considerations have to be made for nonclear cell RCC (ncc-RCC) histotypes.

When considering the PD1 axis as prognostic factor, a study by Abbas et al. on patients undergoing radical renal tumor surgery reported that neither PD-1 positive tumor-infiltrating mononuclear cells or intratumoral PD-L1 expression seemed to significantly impact tumor aggressiveness or clinical outcome in ncc-RCC specimens [166]. Therefore, these cannot be considered good prognosticators.

Although it has been observed that PD-L1 is expressed on tumors of ncc-RCC and sarcomatoid RCC, its role in predicting response to ICI treatments is not clear yet.

Data on the activity of ICIs in patients with ncc-RCC, such as papillary, chromophobe, medullary, Xp11.2 translocation, collecting duct carcinomas, and unclassified carcinomas, or patients with sarcomatoid/rhabdoid differentiation, are limited, due to the diversity of this population and the small numbers in each subset, with consequent low representation in clinical trials.

In a recent pooled analysis on 43 patients with metastatic ncc-RCC or with clear cell cancer with >20% sarcomatoid or rhabdoid differentiation treated with a PD1 or PD-L1 targeting agent (either as monotherapy or in association), the overall response rate was 31% in treatment-naive patients [167]. Patients with RCC with sarcomatoid and/or rhabdoid differentiation and papillary RCC experienced higher ORR, while no patient with chromophobe RCC or unclassified carcinomas responded. The median OS was 12.9 months, and a 12-month OS rate was 64%.

In particular, it has been demonstrated that RCC with a sarcomatoid differentiation may express PD1 and PD-L1 at higher rates than clear cell RCC, and, thus, ICIs have shown initial promising efficacy in this population [168].

A retrospective exploratory analysis from CheckMate 214 showed an impressive efficacy in patients with intermediate/poor risk RCC with sarcomatoid features treated with nivolumab plus ipilimumab versus sunitinib. A higher proportion of patients with sarcomatoid RCC had baseline tumor PD-L1 expression ≥ 1% compared to the CheckMate214 intention-to-treat population (47% vs. 26%, respectively) [15,169].

Furthermore, it has been observed that PD-L1 may play a key role in the biology of Xp11.2 translocation RCC. In fact, in a study by Choueiri, 30% of patients had PD-L1 positivity in tumor cells, and 90% harbored PD-L1+ tumor-infiltrating mononuclear cells. Similarly, in collecting duct carcinoma, 20% of patients expressed PD-L1 on tumor cells [170]. Therefore, these features could represent an important therapeutic target for these histotypes, for which few therapeutic options are currently available.

It is clear that the role of PD-L1 expression as predictive factor is still controversial and under great debate. Further studies are needed to assess whether and how this biomarker will be able to become a valid and robust prognosticator and clinical tool.

## 7. Future Perspectives

In the fast-changing and stimulating scenario of immunotherapy for RCC treatment, several molecules are being investigated as potential targets, beyond the already consolidated PD-1 and CTLA-4. Among the most promising molecules, a mention has to be made on chemokine receptors, the V-domain immunoglobulin-containing suppressor of T-cell activation (VISTA), the soluble lymphocyte-activation gene-3 (LAG-3), OX40 (CD134) and the B and T lymphocyte attenuator (BTLA) [171,172].

In this situation of growing complexity, it seems clear that the sole assessment of PD-1/PD-L1 expression cannot reflect tumor dynamicity.

Besides PD-L1 expression, T-cell density in pretreated samples, T-cell receptor clonality, mutational or neoantigen burden, immunogen signatures, assessment of peripheral T-cell populations, and multiplex IHC with assessment of tumor and immune-cell phenotypes are currently under investigation [173,174]. Combining these strategies in order to evaluate the immune status of the tumor microenvironment will probably result in the identification of more effective biomarkers.

Fundamental will be the identification of different cellular profiles combined with the genetic architecture of RCC, in order to achieve a precision medicine approach and to guide treatment decision among anti-angiogenic agents, ICIs, or a combination of these. For example, VHL and PBRM1 mutant tumors, which have been shown to be associated with both an immune profile and an angiogenic signature, would probably benefit most from a combination treatment [175,176]. In contrast, loss of BAP1, proved to be associated with decreased angiogenic signaling, would probably benefit mostly from ICI treatment [177]. The choice between the combination and sequencing approach will also be essential to improve outcomes for RCC patients.

Therefore, foreseeing the paramount importance of identifying tumor microenvironment changes during the course of different treatments, avoiding the clinical impact of repeated multiple biopsies, the identification of valid biomarkers either on circulating tumor cells or exosomes will become mandatory in the next future [178].

This will be a fundamental process to allow the optimization of treatment choice in cancer patients.

## 8. Conclusions

The interplay between tumor immune status and cancer-related systemic inflammation is crucial for the outcome of RCC treatment with ICIs. However, to date, there is a great unmet need for a more precise characterization of the role of systemic inflammation and inflammatory mediators, both as prognostic and predictive factors of response to these drugs.

It is unquestionably accepted that immunotherapy has profoundly changed the outcomes of patients affected by RCC and is representing an unparalleled revolution. However, in a scenario with several available therapeutic options, future researches will have to shed light on how to best tailor treatments depending on every patient’s specific biological and clinical features.

Due to the availability of multiple options, the sequencing of agents is becoming a daily challenge. The identification of biomarkers to guide therapeutic choices in RCC is advocated. This will help clinicians optimize the selection of treatments, avoiding the exposure to the adverse events related to therapies with predictable low likelihood of clinical benefit, also resulting in better, more cost-effective clinical management.

Due to the paramount importance of the complex interplay among immunological status and immunotherapy, the identification of validated and reproducible biomarkers associated with tumor immune status will be essential to improve the clinical management and, consequently, the outcomes for these patients.

## Figures and Tables

**Figure 1 cancers-11-01935-f001:**
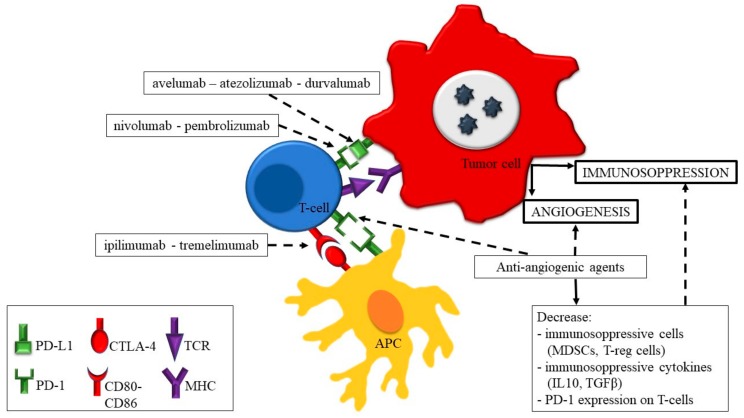
The synergistic effect of anti-angiogenetic agents and immune checkpoint inhibitors on tumor-derived angiogenesis and immunosuppression (dotted line: inhibiting action, continuous line: activating action). Abbreviations: APC = antigen-presenting cell; PD-L1 = programmed death-ligand 1; PD-1; programmed death 1; CTLA-4 = cytotoxic T lymphocyte antigen 4; TCR = T-cell receptor; MHC = major histocompatibility complex; MDSCs = myeloid-derived suppressor cells.

**Table 1 cancers-11-01935-t001:** First-line trials in advanced renal cell carcinoma, combining anti-angiogenic agents and immune checkpoint inhibitors.

Trial Name	Trial Phase	Agents	No. of Patients	mPFS (Months)	Overall Response Rate	Ref.
Checkmate 016	I	Nivolumab + sunitinib	33	48.9	52%	[19]
Checkmate 016	I	Nivolumab + pazopanib	20	31.4	45%	[19]
NCT02133742	Ib	Axitinib + pembrolizumab	52	15.1	71%	[20]
NCT00372853	I	Tremelimumab + sunitinib	28	NA	76%	[21]
KEYNOTE-018	I/II	Pazopanib + pembrolizumab	10	NA	60%	[22]
IMmotion150	II	Bevacizumab + atezolizumab vs. atezolizumab vs. sunitinib	305	11.7	NA	[16]
IMmotion151	III	Bevacizumab + atezolizumab vs. sunitinib	101 (ongoing)	NA	32% (ongoing)	[23]
JAVELIN Renal 100	Ib	Axitinib + avelumab	55	NA	58%	[24]
JAVELIN Renal 101	III	Axitinib + avelumab vs. sunitinib	886	13.8	51%	[17]
CLEAR	III	Lenvatinib + everolimus or pembrolizumab vs. sunitinib	Ongoing	NA	NA	[25]
KEYNOTE-426	III	Axitinib + pembrolizumab vs. sunitinib	Ongoing	NA	NA	[18]
Checkmate 9ER	III	Nivolumab + cabozantinib vs. sunitinib	Ongoing	NA	NA	[26]
COSMIC-313	III	Nivolumab + ipilimumab + cabozantinib vs. nivolumab + ipilimumab + placebo	Ongoing	NA	NA	NA

Abbreviations: mPFS = median progression-free survival, NA = not available, Ref. = reference.

**Table 2 cancers-11-01935-t002:** Characteristics of PD-L1 antibody clones and cutoffs used in phase II and phase III trials evaluating various ICI regimens. Cutoffs are indicated as the percentage of PD-L1 positive cells at immunohistochemistry (where not otherwise indicated).

Trial Name	ICI	PD-L1 Antibody Clone	Developer	Cutoff	Reference
Checkmate 025	Nivolumab	Not reported	Dako	PD-L1 ≥ 1% vs. <1% and ≥5% vs. <5%	[13]
Checkmate 214	Ipilimumab, nivolumab	28-8	Dako	PD-L1 ≥ 1% vs. <1%	[15]
IMMotion 150	Atezolizumab	SP142	Ventana	PD-L1 < 1% or absent (IC 0), ≥1% to <5% (IC 1), ≥5% to <10% (IC 2), or ≥10% (IC3)	[16]
IMMotion 151	Atezolizumab	SP142	Ventana	PD-L1 < 1% vs. ≥1%	[23]
Javelin Renal 101	Avelumab	SP263	Ventana	PD-L1 ≥ 1%	[17]
CLEAR	Pembrolizumab	Not reported	Not reported	Not reported	[25]
KEYNOTE-426	Pembrolizumab	22C3	Agilent Technologies	Combined positive score (PD-L1+ cell no. divided by tumor cell no., multiplied by 100)> or <1	[18]
Checkmate 9ER	Nivolumab	Not reported	Not reported	Not reported	[26]

Abbreviations: ICIs = immune checkpoint inhibitors, PD-L1 = programmed death-ligand 1.

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
