# Peer review of "The Interplay between Inflammation, Anti-Angiogenic Agents, and Immune Checkpoint Inhibitors: Perspectives for Renal Cell Cancer Treatment"

_cancers, 2019, doi:10.3390/cancers11121935_

Round 1

Reviewer 1 Report

The authors extensively review the issue of immunotherapy in renal cell carcinoma. The manuscript is well written and well organized.

I would suggest to add a table summarizing the several clones of PD-L1 with the different cut-off used in each study mentioned by the authors. This would help a better undestanding of the usefulness of PD-L1 as predictive marker.

I would suggest to add a paragraph discussing the role of immunotherapy in the main histotypes.

Reviewer 2 Report

Well written review.

The authors discussed the inflammation in the aspect of prognostic or predictive aspect. Because there are drugs for inflammation such as NSAIDS or anti-IL-1b (canakinumab). Is it possible to comment on the role of these agents for the treatment of RCC?
